# Recurrent Transformer Networks for Semantic Correspondence

**Seungryong Kim[1], Stephen Lin[2], Sangryul Jeon[1], Dongbo Min[3], and Kwanghoon Sohn[1],***

[1]Yonsei University, Seoul, South Korea, [2]Microsoft Research, Beijing, China,
[3]Ewha Womans University, Seoul, South Korea
{srkim89,cheonjsr,khsohn}@yonsei.ac.kr, stevelin@microsoft.com,
dbmin@ewha.ac.kr

## Abstract

We present recurrent transformer networks (RTNs) for obtaining dense correspondences between semantically similar images. Our networks accomplish this through an iterative process of estimating spatial transformations between the input images and using these transformations to generate aligned convolutional activations. By directly estimating the transformations between an image pair, rather than employing spatial transformer networks to independently normalize each individual image, we show that greater accuracy can be achieved. This process is conducted in a recursive manner to refine both the transformation estimates and the feature representations. In addition, a technique is presented for weakly-supervised training of RTNs that is based on a proposed classification loss. With RTNs, state-of-the-art performance is attained on several benchmarks for semantic correspondence.

## 1 Introduction

Establishing dense correspondences across *semantically* similar images can facilitate a variety of computer vision applications including non-parametric scene parsing, semantic segmentation, object detection, and image editing [25, 22, 19]. In this semantic correspondence task, the images resemble each other in content but differ in object appearance and configuration, as exemplified in the images with different car models in Fig. 1(a-b). Unlike the dense correspondence computed for estimating depth [34] or optical flow [4], semantic correspondence poses additional challenges due to intra-class appearance and shape variations among different instances from the same object or scene category.

To address these challenges, state-of-the-art methods generally extract deep convolutional neural network (CNN) based descriptors [5, 44, 18], which provide some robustness to appearance variations, and then perform a regularization step to estimate spatially regularized geometric fields. The most recent techniques handle geometric deformations in addition to appearance variations within deep CNNs. These methods can generally be classified into two categories, namely methods for geometric invariance in the feature extraction step, e.g., spatial transformer networks (STNs) [15, 5, 19], and methods for geometric invariance in the regularization step, e.g., geometric matching networks [30, 31]. The STN-based methods infer geometric deformation fields within a deep network and transform the convolutional activations to provide geometric-invariant features [5, 41, 19]. While this approach has shown geometric invariance to some extent, we conjecture that directly estimating the geometric deformations between a pair of input images would be more robust and precise than learning to transform each individual image to a geometric-invariant feature representation. This direct estimation approach is used by geometric matching-based techniques [30, 31], which recover a matching model directly through deep networks. Drawbacks of these methods include that globally-varying geometric fields are inferred, and only fixed, untransformed versions of the features are used.

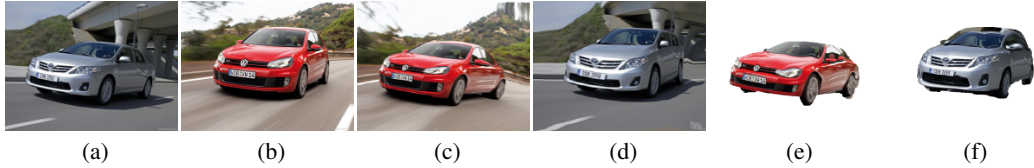

|  (a)  |  (b)  |  (c)  |  (d)  |  (e)  |  (f)  |

Figure 1: Visualization of results from RTNs: (a) source image; (b) target image; (c), (d) warped source and target images using dense correspondences from RTNs; (e), (f) pseudo ground-truth transformations in [36]. Our RTNs learn to infer transformations without ground-truth supervision.

In this paper, we present recurrent transformer networks (RTNs) for overcoming the aforementioned limitations of current semantic correspondence techniques. As illustrated in Fig. 2, the key idea of RTNs is to directly estimate the geometric transformation fields between two input images, like what is done by geometric matching-based approaches [30, 31], but also apply the estimated field to transform the convolutional activations of one of the images, similar to STN-based methods [15, 5, 19]. We additionally formulate the RTNs to recursively estimate the geometric transformations, which are used for iterative geometric alignment of feature activations. In this way, regularization is enhanced through recursive refinement, while feature extraction is likewise iteratively refined according to the geometric transformations as well as jointly learned with the regularization. Moreover, the networks are learned in a weakly-supervised manner via a proposed classification loss defined between the source image features and the geometrically-aligned target image features, such that the correct transformation is identified by the highest matching score while other transformations are considered as negative examples.

The presented approach is evaluated on several common benchmarks and examined in an ablation study. The experimental results show that this model outperforms the latest weakly-supervised and even supervised methods for semantic correspondence.

## 2 Related Work

**Semantic Correspondence**    To elevate matching quality, most conventional methods for semantic correspondence focus on improving regularization techniques while employing handcrafted features such as SIFT [27]. Liu et al. [25] pioneered the idea of dense correspondence across different scenes, and proposed SIFT flow. Inspired by this, methods have been presented based on deformable spatial pyramids (DSP) [17], object-aware hierarchical graphs [39], exemplar LDA [3], joint image set alignment [45], and joint co-segmentation [36]. As all of these techniques use handcrafted descriptors and regularization methods, they lack robustness to geometric deformations.

Recently, deep CNN-based methods have been used in semantic correspondence as their descriptors provide some degree of invariance to appearance and shape variations. Among them are techniques that utilize a 3-D CAD model for supervision [44], employ fully convolutional feature learning [5], learn filters with geometrically consistent responses across different object instances [28], learn networks using dense equivariant image labelling [37], exploit local self-similarity within a fully convolutional network [18, 19], and estimate correspondences using object proposals [7, 8, 38]. However, none of these methods is able to handle non-rigid geometric variations, and most of them are formulated with handcrafted regularization. More recently, Han et al. [9] formulated the regularization into the CNN but do not deal explicitly with the significant geometric variations encountered in semantic correspondence.

**Spatial Invariance**    Some methods aim to alleviate spatial variation problems in semantic correspondence through extensions of SIFT flow, including scale-less SIFT flow (SLS) [11], scale-space SIFT flow (SSF) [29], and generalized DSP (GDSP) [13]. A generalized PatchMatch algorithm [1] was proposed for efficient matching that leverages a randomized search scheme. It was utilized by HaCohen et al. [6] in a non-rigid dense correspondence (NRDC) algorithm. Spatial invariance to scale and rotation is provided by DAISY filter flow (DFF) [40]. While these aforementioned techniques provide some degree of geometric invariance, none of them can deal with affine transformations over an image. Recently, Kim et al. [20, 21] proposed the discrete-continuous transformation matching

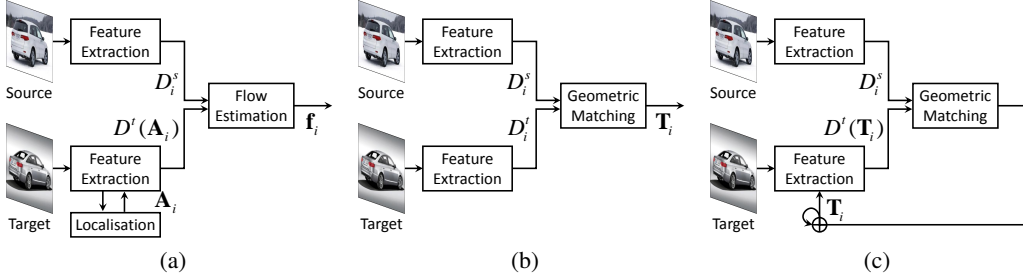

Figure 2: Intuition of RTNs: (a) methods for geometric invariance in the feature extraction step, e.g., STN-based methods [5, 19], (b) methods for geometric invariance in the regularization step, e.g., geometric matching-based methods [30, 31], and (c) RTNs, which weave the advantages of both existing STN-based methods and geometric matching techniques, by recursively estimating geometric transformation residuals using geometry-aligned feature activations.

(DCTM) framework where dense affine transformation fields are inferred using a hand-designed energy function and regularization.

To deal with geometric variations within CNNs, STNs [15] offer a way to provide geometric invariance by warping features through a global transformation. Inspired by STNs, Lin et al. [23] proposed inverse compositional STNs (IC-STNs) that replaces the feature warping with transformation parameter propagation. Kanazawa et al. [16] presented WarpNet that predicts a warp for establishing correspondences. Rocco et al. [30, 31] proposed a CNN architecture for estimating a geometric matching model for semantic correspondence. However, they estimate only globally-varying geometric fields, thus leading to limited performance in dealing with locally-varying geometric deformations. To deal with locally-varying geometric variations, some methods such as UCN-spatial transformer (UCN-ST) [5] and convolutional affine transformer-FCSS (CAT-FCSS) [19] employ STNs [15] at the pixel level. Similarly, Yi et al. [41] proposed the learned invariant feature transform (LIFT) to learn sparsely, locally-varying geometric fields, inspired by [42]. However, these methods determine geometric fields by accounting for the source and target images independently, rather than jointly, which limits their prediction ability.

## 3  Background

Let us denote *semantically* similar source and target images as $I^s$ and $I^t$, respectively. The objective is to establish a correspondence field $\mathbf{f}_i = [u_i, v_i]^T$ between the two images that is defined for each pixel $i = [i_{\mathbf{x}}, i_{\mathbf{y}}]^T$ in $I^s$. Formally, this involves first extracting handcrafted or deep features, denoted by $D_i^s$ and $D_i^t$, from $I^s$ and $I^t$ within local receptive fields, and then estimating the correspondence field $\mathbf{f}_i$ of the source image by maximizing the feature similarity between $D_i^s$ and $D_{i+\mathbf{f}_i}^t$ over a set of transformations using handcrafted or deep geometric regularization models. Several approaches [25, 18] assume the transformation to be a 2-D translation with negligible variation within local receptive fields. As a result, they often fail to handle complicated deformations caused by scale, rotation, or skew that may exist among object instances. For greater geometric invariance, recent approaches [20, 21] have modeled the deformations as an affine transformation field represented by a $2 \times 3$ matrix

$$\mathbf{T}_i = [\mathbf{A}_i, \mathbf{f}_i] \tag{1}$$

that maps pixel $i$ to $i' = i + \mathbf{f}_i$. Specifically, they maximize the similarity between the source $D_i^s$ and target $D_{i'}^t(\mathbf{A}_i)$, where $D(\mathbf{A}_i)$ represents the feature extracted from spatially-varying local receptive fields transformed by a $2 \times 2$ matrix $\mathbf{A}_i$ [5, 19]. For simplicity, we denote $D^t(\mathbf{T}_i) = D_{i+\mathbf{f}_i}^t(\mathbf{A}_i)$.

Approaches for geometric invariance in semantic correspondence can generally be classified into two categories. The first group infers the geometric fields in the feature extraction step by minimizing a matching objective function [5, 19], as exemplified in Fig. 2(a). Concretely, $\mathbf{A}_i$ is learned without a ground-truth $\mathbf{A}_i^*$ by minimizing the difference between $D_i^s$ and $D_{i+\mathbf{f}_i}^t(\mathbf{A}_i)$ according to a ground-truth flow field $\mathbf{f}_i^*$. This enables explicit feature learning which aims to minimize/maximize convolutional activation differences between matching/non-matching pixel pairs [5, 19]. However, ground-truth flow fields $\mathbf{f}_i^*$ are still needed for learning the networks, and it predicts the geometric

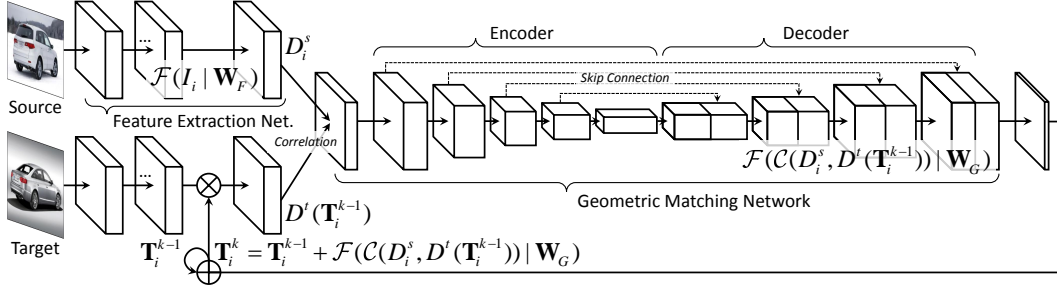

Figure 3: Network configuration of RTNs, consisting of a feature extraction network and a geometric matching network in a recurrent structure.

fields $\mathbf{A}_i$ based only on the source or target feature, without jointly considering the source and target, thus limiting performance.

The second group estimates a geometric matching model directly through deep networks by considering the source and target features simultaneously [30, 31]. These methods formulate the geometric matching networks by mimicking conventional RANSAC-like methods [14] through feature extraction and geometric matching steps. As illustrated in Fig. 2(b), the geometric fields $\mathbf{T}_i$ are predicted in a feed-forward network from extracted source features $D_i^s$ and target features $D_i^t$. By learning to extract source and target features and predict geometric fields in an end-to-end manner, more robust geometric fields can be estimated compared to existing STN-based methods that consider source or target features independently as shown in [31]. A major limitation of these learning-based methods is the lack of ground-truth geometric fields $\mathbf{T}_i^*$ between source and target images. To alleviate this problem, some methods use self-supervision such as synthetic transformations [30] or weak-supervision such as soft-inlier maximization [31], but these approaches constrain the global geometric field only. Moreover, these methods utilize feature descriptors extracted from the original upright images, rather than from geometrically transformed images, which limits their capability to represent severe geometric variations.

## 4 Recurrent Transformer Networks

### 4.1 Motivation and Overview

In this section, we describe the formulation of recurrent transformer networks (RTNs). The objective of our networks is to learn and infer locally-varying affine deformation fields $\mathbf{T}_i$ in an end-to-end and weakly-supervised fashion using only image pairs without ground-truth transformations $\mathbf{T}_i^*$. Toward this end, we present an effective and efficient integration of the two existing approaches for geometric invariance, i.e., STN-based feature extraction networks [5, 19] and geometric matching networks [30, 31], that includes a novel weakly-supervised loss function tailored to our objective. Specifically, the final geometric field is recursively estimated by deforming the activations of feature extraction networks according to the intermediate output of the geometric matching networks, in contrast to existing approaches based on geometric matching which consider only fixed, upright versions of features [30, 31]. At the same time, our method outperforms STN-based approaches [5, 19] by using a deep CNN-based geometric matching network instead of handcrafted matching criteria. Our recurrent geometric matching approach intelligently weaves the advantages of both existing STN-based methods and geometric matching techniques, by recursively estimating geometric transformation residuals using geometry-aligned feature activations.

Concretely, our networks are split into two parts, as shown in Fig. 3: a *feature extraction network* to extract source $D_i^s$ and target $D^t(\mathbf{T}_i)$ features, and a *geometric matching network* to infer the geometric fields $\mathbf{T}_i$. To learn these networks in a weakly-supervised manner, we formulate a novel classification loss defined without ground-truth $\mathbf{T}_i^*$ based on the assumption that the transformation which maximizes the similarity of the source features $D_i^s$ and transformed target features $D^t(\mathbf{T}_i)$ at a pixel $i$ should be correct, while the matching scores of other transformation candidates should be minimized.

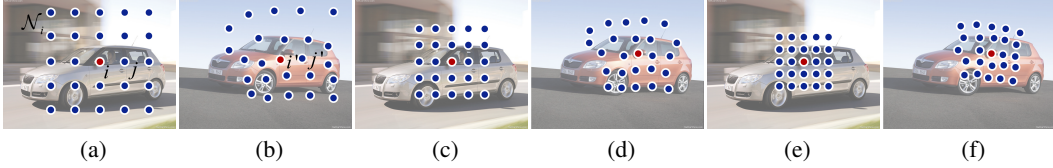

Figure 4: Visualization of search window $\mathcal{N}_i$ in RTNs (e.g., $|\mathcal{N}_i| : 5 \times 5$): Source images with the search window of (a) stride 4, (c) stride 2 , (e) stride 1, and target images with (b), (d), (f) transformed points for (a), (c), (e), respectively. As evolving iterations, the dilate strides are reduced to consider precise matching details.

## 4.2  Feature Extraction Network

To extract convolutional features for source $D^s$ and target $D^t$, the input source and target images ($I^s$, $I^t$) are first passed through fully-convolutional feature extraction networks with shared parameters $\mathbf{W}_F$ such that $D_i = \mathcal{F}(I_i|\mathbf{W}_F)$, and the feature for each pixel then undergoes $L_2$ normalization. In the recurrent formulation, at each iteration the target features $D^t$ can be extracted according to $\mathbf{T}_i$ such that $D^t(\mathbf{T}_i) = \mathcal{F}(I^t(\mathbf{T}_i)|\mathbf{W}_F)$. However, extracting each feature by transforming local receptive fields within the target image $I^t$ according to $\mathbf{T}_i$ for each pixel $i$ and then passing it through the networks would be time-consuming when iterating the networks. Instead, we employ a strategy similar to UCN-ST [5] and CAT-FCSS [19] by first extracting the convolutional features of the entire image $I^t$ by passing it through the networks except for the last convolutional layer, and then computing $D^t(\mathbf{T}_i)$ by transforming the resultant convolutional features and finally passing it through the last convolution with stride to combine the transformed activations independently [5, 19]. It should be noted that any convolutional features [35, 12, 19] could be used in this framework. In experiments, we used CAT-FCSS [19], and sampled activations after pooling layers such as 'conv4-4' for VGGNet [35] and 'conv4-23' for ResNet [12].

## 4.3  Recurrent Geometric Matching Network

**Constraint Correlation Volume**  To predict the geometric fields from two convolutional features $D^s$ and $D^t$, we first compute the correlation volume with respect to translational motion only [30, 31] and then pass it to subsequent convolutional layers to determine dense affine transformation fields. As shown in [31], this two-step approach reliably prunes incorrect matches. Specifically, the similarity between two extracted features is computed as the cosine similarity with $L_2$ normalization:

$$\mathcal{C}(D_i^s, D^t(\mathbf{T}_j)) = <D_i^s, D^t(\mathbf{T}_j)>/\sqrt{\sum_l <D_i^s, D^t(\mathbf{T}_l)>^2}, \tag{2}$$

where $j, l \in \mathcal{N}_i$ for the search window $\mathcal{N}_i$ of pixel $i$.

Compared to [30, 31] that consider all possible samples within an image, the constraint correlation volume defined within $\mathcal{N}_i$ reduces the matching ambiguity and computational times. However, due to the limited search window range, it may not cover large geometric variations. To alleviate this limitation, inspired by [43], we utilize dilation techniques in a manner that the local neighborhood $\mathcal{N}_i$ is enlarged with larger stride than 1 pixel, and this dilation is reduced as the iterations progress, as exemplified in Fig. 4.

**Recurrent Geometry Estimation**  Based on this matching similarity, the recurrent geometry estimation network with parameters $\mathbf{W}_G$ iteratively estimates the residual between the previous and current geometric transformation fields as

$$\mathbf{T}_i^k - \mathbf{T}_i^{k-1} = \mathcal{F}(\mathcal{C}(D_i^s, D^t(\mathbf{T}_i^{k-1}))|\mathbf{W}_G), \tag{3}$$

where $\mathbf{T}_i^k$ denotes the transformation fields at the $k$-th iteration. The final geometric fields are then estimated in a recurrent manner as follows:

$$\mathbf{T}_i = \mathbf{T}_i^0 + \sum_{k\in\{1,..,K_{\max}\}} \mathcal{F}(\mathcal{C}(D_i^s, D^t(\mathbf{T}_i^{k-1}))|\mathbf{W}_G), \tag{4}$$

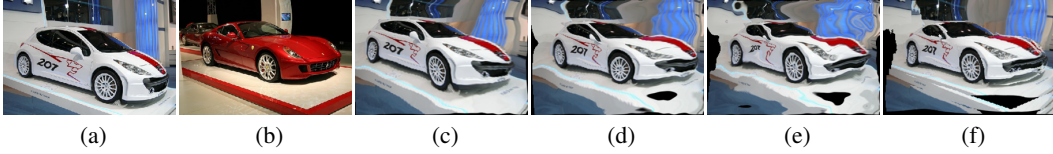

<div style="text-align:center">(a)      (b)      (c)      (d)      (e)      (f)</div>

Figure 5: Convergence of RTNs: (a) source image; (b) target image; Iterative evolution of warped images (c), (d), (e), and (f) after iteration 1, 2, 3, and 4. In the recurrent formulation of RTNs, the predicted transformation field becomes progressively more accurate through iterative estimation.

where $K_{\max}$ denotes the maximum iteration and $\mathbf{T}_i^0$ is an initial geometric field. Unlike [30, 31] which estimate a global affine or thin-plate spline transformation field, we formulate the encoder-decoder networks as in [32] to estimate locally-varying geometric fields. Moreover, our networks are formulated in a fully-convolutional manner, thus source and target inputs of any size can be processed, in contrast to [30, 31] which can take inputs of only a fixed size.

Iteratively inferring affine transformation residuals boosts matching precision and facilitates convergence. Moreover, inferring residuals instead of carrying the input information through the network has been shown to improve network optimization [12]. As shown in Fig. 5, the predicted transformation field becomes progressively more accurate through iterative estimation.

## 4.4 Weakly-supervised Learning

A major challenge of semantic correspondence with deep CNNs is the lack of ground-truth correspondence maps for training. Obtaining such ground-truth data through manual annotation is labor-intensive and may be degraded by subjectivity [36, 7, 8]. To learn the networks using only weak supervision in the form of image pairs, we formulate the loss function based on the intuition that the matching score between the source feature $D_i^s$ at each pixel $i$ and the target feature $D^t(\mathbf{T}_i)$ should be maximized while keeping the scores of other transformation candidates low. This can be treated as a classification problem in that the network can learn a geometric field as a hidden variable for maximizing the scores for matchable $\mathbf{T}_i$ while minimizing the scores for non-matchable transformation candidates. The optimal fields $\mathbf{T}_i$ can be learned with a classification loss [19] in a weakly-supervised manner by minimizing the energy function

$$\mathcal{L}(D_i^s, D^t(\mathbf{T})) = -\sum_{j \in \mathcal{M}_i} p_j^* \log(p(D_i^s, D^t(\mathbf{T}_j))), \tag{5}$$

where the function $p(D_i^s, D^t(\mathbf{T}_j))$ is a softmax probability defined as

$$p(D_i^s, D^t(\mathbf{T}_j)) = \frac{\exp(\mathcal{C}(D_i^s, D^t(\mathbf{T}_j)))}{\sum_{l \in \mathcal{M}_i} \exp(\mathcal{C}(D_i^s, D^t(\mathbf{T}_l)))}, \tag{6}$$

with $p_j^*$ denoting a class label defined as 1 if $j = i$, and 0 otherwise for $j \in \mathcal{M}_i$ for the search window $\mathcal{M}_i$, such that the center point $i$ within $\mathcal{M}_i$ becomes a positive sample while the other points are negative samples.

With this loss function, the derivatives $\partial \mathcal{L}/\partial D^s$ and $\partial \mathcal{L}/\partial D^t(\mathbf{T})$ of the loss function $\mathcal{L}$ with respect to the features $D^s$ and $D^t(\mathbf{T})$ can be back-propagated into the feature extraction networks $\mathcal{F}(\cdot|\mathbf{W}_F)$. Explicit feature learning in this manner with the classification loss has been shown to be reliable [19]. Likewise, the derivatives $\partial \mathcal{L}/\partial D^t(\mathbf{T})$ and $\partial D^t(\mathbf{T})/\partial \mathbf{T}$ of the loss function $\mathcal{L}$ with respect to geometric fields $\mathbf{T}$ can be back-propagated into the geometric matching networks $\mathcal{F}(\cdot|\mathbf{W}_G)$ to learn these networks without ground truth $\mathbf{T}^*$.

It should be noted that our loss function is conceptually similar to [31] in that it is formulated with source and target features in a weakly-supervised manner. While [31] utilizes only positive samples in learning feature extraction networks, our method considers both positive and negative samples to enhance network training.

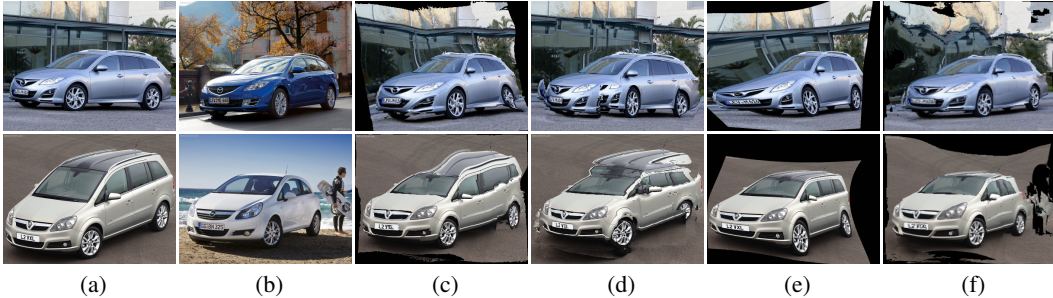

|   (a)   |   (b)   |   (c)   |   (d)   |   (e)   |   (f)   |

Figure 7: Qualitative results on the TSS benchmark [36]: (a) source image, (b) target image, (c) DCTM [18], (d) SCNet [9], (e) GMat. w/Inl. [31], and (f) RTNs. The source images are warped to the target images using correspondences.

## 5 Experimental Results and Discussion

### 5.1 Experimental Settings

In the following, we comprehensively evaluated our RTNs through comparisons to state-of-the-art methods for semantic correspondence, including SF [25], DSP [17], Zhou et al. [44], Taniai et al. [36], PF [7], SCNet [9], DCTM [18], geometric matching (GMat.) [30], and GMat. w/Inl. [31], as well as employing the SIFT flow optimizer[1] together with UCN-ST [5], FCSS [18], and CAT-FCSS [19]. Performance was measured on the TSS dataset [36], PF-WILLOW dataset [7], and PF-PASCAL dataset [8]. In Sec. 5.2, we first analyze the effects of the components within RTNs, and then evaluate matching results with various benchmarks and quantitative measures in Sec. 5.3.

### 5.2 Ablation Study

To validate the components within RTNs, we evaluated the matching accuracy for different numbers of iterations, with various window sizes of $\mathcal{N}_i$, for different backbone feature extraction networks such as VGGNet [35], CAT-FCSS [19], and ResNet [12], and with pretrained or learned backbone networks. For quantitative assessment, we examined the matching accuracy on the TSS benchmark [36], as described in the following section. As shown in Fig. 6, RTNs w/ResNet [12] converge in $3-5$ iterations. By enlarging the window size of $\mathcal{N}_i$, the matching accuracy improves until $9\times9$ with longer training and testing times, but larger window sizes reduce matching accuracy due to greater matching ambiguity. Note that $\mathcal{M}_i = \mathcal{N}_i$. Table 1 shows that among the many state-of-the-art feature extraction networks, ResNet [12] exhibits the best performance for our approach. As shown in comparisons between pretrained and learned backbone networks, learning the feature extraction networks jointly with geometric matching networks can boost matching accuracy, as similarly seen in [31].

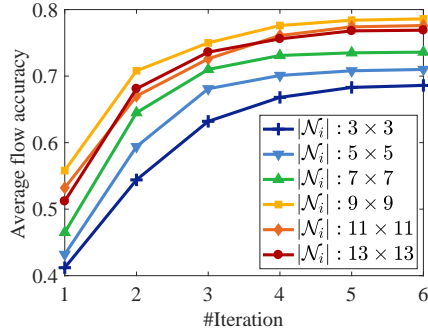

Figure 6: Convergence analysis of RTNs w/ResNet [12] for various numbers of iterations and search window sizes on the TSS benchmark [36].

### 5.3 Matching Results

**TSS Benchmark** We evaluated RTNs on the TSS benchmark [36], which consists of 400 image pairs divided into three groups: FG3DCar [24], JODS [33], and PASCAL [10]. As in [18, 20], flow accuracy was measured by computing the proportion of foreground pixels with an absolute flow endpoint error that is smaller than a threshold of $T = 5$, after resizing each image so that

| Methods | Feature | Regular. | Superv. | FG3D. | JODS | PASC. | Avg. |
|---|---|---|---|---|---|---|---|
| SF [25] | SIFT | SF | - | 0.632 | 0.509 | 0.360 | 0.500 |
| DSP [17] | SIFT | DSP | - | 0.487 | 0.465 | 0.382 | 0.445 |
| Taniai et al. [36] | HOG | TSS | - | 0.830 | 0.595 | 0.483 | 0.636 |
| PF [7] | HOG | LOM | - | 0.786 | 0.653 | 0.531 | 0.657 |
| DCTM [18] | CAT-FCSS$^\dagger$ | DCTM | - | 0.891 | 0.721 | 0.610 | 0.740 |
| UCN-ST [5] | UCN-ST | SF | Sup. | 0.853 | 0.672 | 0.511 | 0.679 |
| FCSS [18, 19] | FCSS | SF | Weak. | 0.832 | 0.662 | 0.512 | 0.668 |
| | CAT-FCSS | SF | Weak. | 0.858 | 0.680 | 0.522 | 0.687 |
| SCNet [9] | VGGNet | AG | Sup. | 0.764 | 0.600 | 0.463 | 0.609 |
| | VGGNet | AG+ | Sup. | 0.776 | 0.608 | 0.474 | 0.619 |
| GMat. [30] | VGGNet | GMat. | Self. | 0.835 | 0.656 | 0.527 | 0.673 |
| | ResNet | GMat. | Self. | 0.886 | 0.758 | 0.560 | 0.735 |
| GMat. w/Inl. [31] | ResNet | GMat. | Weak. | 0.892 | 0.758 | 0.562 | 0.737 |
| RTNs | VGGNet$^\dagger$ | R-GMat. | Weak. | 0.875 | 0.736 | 0.586 | 0.732 |
| RTNs | VGGNet | R-GMat. | Weak. | 0.893 | 0.762 | 0.591 | 0.749 |
| RTNs | CAT-FCSS | R-GMat. | Weak. | 0.889 | 0.775 | 0.611 | 0.758 |
| RTNs | ResNet | R-GMat. | Weak. | **0.901** | **0.782** | **0.633** | **0.772** |

Table 1: Matching accuracy compared to state-of-the-art correspondence techniques (with feature, regularization, and supervision) on the TSS benchmark [36]. $^\dagger$ denotes a pre-trained feature.

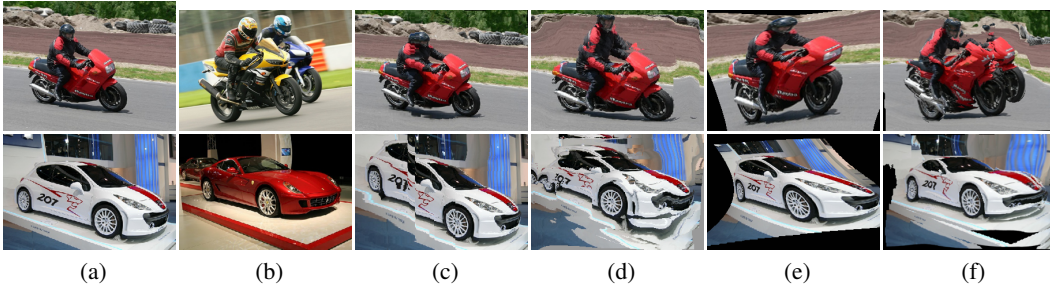

| (a) | (b) | (c) | (d) | (e) | (f) |

Figure 8: Qualitative results on the PF-WILLOW benchmark [7]: (a) source image, (b) target image, (c) UCN-ST [5], (d) SCNet [9], (e) GMat. w/Inl. [31], and (f) RTNs. The source images are warped to the target images using correspondences.

its larger dimension is 100 pixels. Table 1 compares the matching accuracy of RTNs to state-of-the-art correspondence techniques, and Fig. 7 shows qualitative results. Compared to handcrafted methods [25, 17, 36, 7], most CNN-based methods have better performance. In particular, methods that use STN-based feature transformations, namely UCN-ST [5] and CAT-FCSS [19], show improved ability to deal with geometric variations. In comparison to the geometric matching-based methods GMat. [30] and GMat. w/Inl. [30], RTNs consisting of feature extraction with ResNet and recurrent geometric matching modules provide higher performance. RTNs additionally outperform existing CNN-based methods trained with supervision of flow fields. It should be noted that GMat. w/Inl. [31] was learned with the initial network parameters set through self-supervised learning as in [30]. RTNs instead start from fully-randomized parameters in geometric matching networks.

**PF-WILLOW Benchmark** We also evaluated our method on the PF-WILLOW benchmark [7], which includes 10 object sub-classes with 10 keypoint annotations for each image. For the evaluation metric, we use the probability of correct keypoint (PCK) between flow-warped keypoints and the ground truth [26, 7] as in the experiments of [18, 9, 19]. Table 2 compares the PCK values of RTNs to state-of-the-art correspondence techniques, and Fig. 8 shows qualitative results. Our RTNs exhibit performance competitive to the state-of-the-art correspondence techniques including the latest weakly-supervised and even supervised methods for semantic correspondence. Since RTNs estimate locally-varying geometric fields, it provides more precise localization ability, as shown in the

| Methods | PF-WILLOW [7] | | | PF-PASCAL [8] | | |
|---|---|---|---|---|---|---|
| | $\alpha = 0.05$ | $\alpha = 0.1$ | $\alpha = 0.15$ | $\alpha = 0.05$ | $\alpha = 0.1$ | $\alpha = 0.15$ |
| PF [7] | 0.284 | 0.568 | 0.682 | 0.314 | 0.625 | 0.795 |
| DCTM [18] | 0.381 | 0.610 | 0.721 | 0.342 | 0.696 | 0.802 |
| UCN-ST [5] | 0.241 | 0.540 | 0.665 | 0.299 | 0.556 | 0.740 |
| CAT-FCSS [19] | 0.362 | 0.546 | 0.692 | 0.336 | 0.689 | 0.792 |
| SCNet [9] | 0.386 | 0.704 | 0.853 | 0.362 | 0.722 | 0.820 |
| GMat. [30] | 0.369 | 0.692 | 0.778 | 0.410 | 0.695 | 0.804 |
| GMat. w/Inl. [31] | 0.370 | 0.702 | 0.799 | 0.490 | 0.748 | 0.840 |
| RTNs w/VGGNet | 0.402 | 0.707 | 0.842 | 0.506 | 0.743 | 0.836 |
| RTNs w/ResNet | **0.413** | **0.719** | **0.862** | **0.552** | **0.759** | **0.852** |

Table 2: Matching accuracy compared to state-of-the-art correspondence techniques on the PF-WILLOW benchmark [7] and PF-PASCAL benchmark [8].

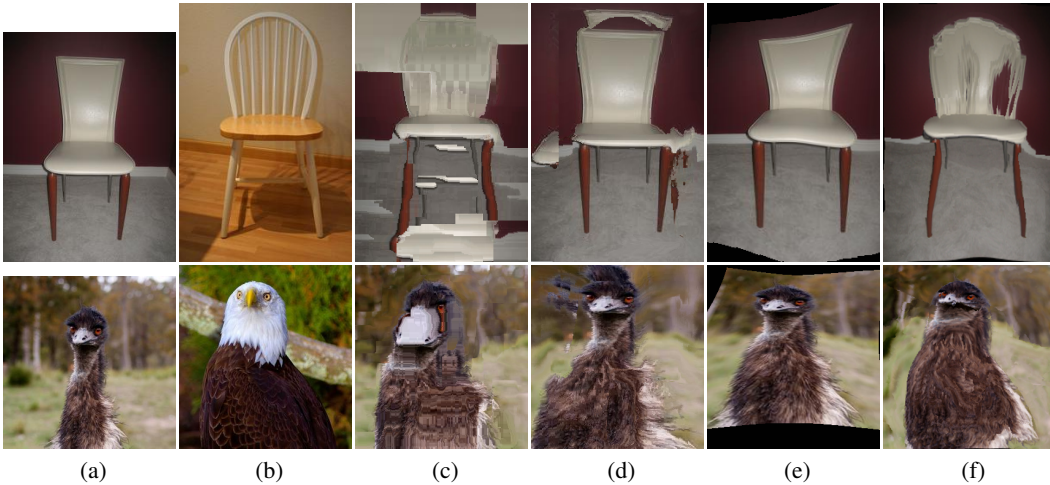

(a)       (b)       (c)       (d)       (e)       (f)

Figure 9: Qualitative results on the PF-PASCAL benchmark [8]: (a) source image, (b) target image, (c) CAT-FCSS w/SF [19], (d) SCNet [9], (e) GMat. w/Inl. [31], and (f) RTNs. The source images are warped to the target images using correspondences.

results of $\alpha = 0.05$, in comparison to existing geometric matching networks [30, 31] which estimate globally-varying geometric fields only.

**PF-PASCAL Benchmark**    Lastly, we evaluated our method on the PF-PASCAL benchmark [8], which contains 1,351 image pairs over 20 object categories with PASCAL keypoint annotations [2]. Following the split in [9, 31], we used 700 training pairs, 300 validation pairs, and 300 testing pairs. For the evaluation metric, we use the PCK between flow-warped keypoints and the ground truth as done in the experiments of [9]. Table 2 summarizes the PCK values, and Fig. 9 shows qualitative results. Similar to the experiments on the PF-WILLOW benchmark [7], CNN-based methods [9, 30, 31] including our RTNs yield better performance, with RTNs providing the highest matching accuracy.

# 6   Conclusion

We presented RTNs, which learn to infer locally-varying geometric fields for semantic correspondence in an end-to-end and weakly-supervised fashion. The key idea of this approach is to utilize and iteratively refine the transformations and convolutional activations via geometric matching between the input image pair. In addition, a technique is presented for weakly-supervised training of RTNs. A direction for further study is to examine how the semantic correspondence of RTNs could benefit single-image 3-D reconstruction and instance-level object detection and segmentation.

## Acknowledgments

This research was supported by Next-Generation Information Computing Development Program through the National Research Foundation of Korea (NRF) funded by the Ministry of Science and ICT (NRF-2017M3C4A7069370).

## Footnotes

[1]For these experiments, we utilized the hierarchical dual-layer belief propagation of SIFT flow [25] together with alternative dense descriptors.

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
