[Reviews · NeurIPS 2018]

Reviewer 1



The paper proposes an recurrent iterative refinement approach for weakly supervised semantic correspondence learning between similar objects. The idea is to predict a flow field transformation for each pixel, such that after transforming the image the source and target images are aligned. This paper builds on the weak-supervision similar to [18] and [29] but predicts the flow, and adds an iterative component where in each step the network outputs the residual to that brings the source and the transformed target images closer. The objective is similar to the concurrent work of [29] (but with negatives) where the idea that transformed pixel j should have the highest feature correlation with the feature at the pixel it got transformed to than any of its immediate neighbors. The paper also proposes a dilated search window for this neighbor radius, where the search window shrinks as the iteration continues such that more finer alignment may be achieved. The paper evaluates against standard benchmarks, improves upon recent state of the art methods, and obtains visually pleasant results. Pros of this paper is that: - Results look good - Doing matching in an iterative style makes sense - Progressively shrinking search window makes sense - In a sense it's a simple approach that brings together a lot of the recent advancements in this field. - Flow rather than affine or TPS allows for finer detail matching. Weaknesses are: In terms of originality & novelty, while the idea to do iteration has yet to be tried in image matching, this idea has been demonstarted in various other applications: - Joao Carreira, Pulkit Agrawal, Katerina Fragkiadaki, Jitendra Malik, "Human Pose Estimation with Iterative Error Feedback", CVPR 2016 - Markus Oberweger, Paul Wohlhart, Vincent Lepetit, "Training a feedback loop for hand pose estimation" ICCV 2015 - Piotr Dollar, Peter Wlinder, Pietro Perona, "Cascaded pose regression", CVPR 2010 The paper should be more generous in citing these papers. In an essence this paper could be seen as a combination of [18] and [29] but with flow and iteration + other tricks. Therefore it's not as original, however, [29] is concurrent and given the quality of results I'm inclined towards acceptance. At the same time, this kind of paper might be more suitable for vision conferences like CVPR/ICCV/ECCV. Authors might want to cite these other related works: - Thewlis et al. Unsupervised learning of object frames by dense equivariant image labelling. NIPS 2017 Does not learn matching transformation but learns a matchable features, also from synthetic deformations - Kanazawa et al. "WarpNet: Weakly Supervised Matching for Single-view Reconstruction", one of the first papers to solve for TPS to match two images, trained in a self-supervised manner through synthetic transformation (line 123) and also they demonstrate the matching results for single-view 3D application, relevant to the conclusion of this paper. Post rebuttal: I stand by my original rating.

Reviewer 2



This paper proposes a recurrent transformer network (RTN) for semantic correspondence that aligns semantically similar images. The main idea is to perform an alternating process of estimating local spatial transformations between the input images and using these transformations to generate better aligned features for further estimations. This iterative residual regression gradually improves the dense alignment between the images. The training is done in weakly-supervised manner by maximizing the score of the estimated transformation relative to other transformations. + The proposed method is clearly motivated and novel. + It combines the recent methods and improves them in an interesting direction. + The weakly-supervised training is adopted with a new loss function. + Experimental results demonstrate its state-of-the-art methods on standard benchmarks. - The introduction is misleading and confusing in some aspects (see below). - Some details are missing in the manuscript (see below). I like the contributions of this paper overall. But, the exposition of this paper needs to be revised, in particular, in the introduction. The authors divide recent methods with geometric deformation into two classes: STN-based [5,18] vs. geometric matching [28,29]. Then, is it adequate to say “most of the state-of-the-art techniques take this approach (STN-based)” given Table 1? And, how about framing them as 'modeling local deformation vs. global deformation' based on line 78-89? To deliver better understanding to general readers, 'STN' sounds too specific and 'geometric matching' sounds ambiguous for the introduction. I highly recommend revising the introduction considering this. Line 31-35 is totally mistaken. The work of [28,29] does not require supervision: [28] is self-supervised and [29] is weakly-supervised. Also, feature representations of [28,29] are not learned independently from transformation learning. Actually, their feature representations are learned jointly end-to-end in [28,29]. Please check them out. In addition, the results of [29] in this submission is different from the CVPR paper version. Please check the final version: http://openaccess.thecvf.com/content_cvpr_2018/papers/Rocco_End-to-End_Weakly-Supervised_Semantic_CVPR_2018_paper.pdf In section 4.4, the set M_i is not properly defined. Each position i has its affine transformation and only the translation factor f_i looks used for final output. Then, is the affine transformation really necessary for better performance? It would be good to show the effect in the ablation study. How is the running time of the proposed method compared to other methods? The proposed method looks able to be trained with strong supervision if available. How is the performance trained with strong supervision from the benchmark datasets?

Reviewer 3



The paper presents a method for establishing dense correspondences between a pair of images. Rather than employing a neural network to generate the output flow in one pass, the paper uses a recurrent neural network to achieve the goal. The motivation comes from iterative methods for non-rigid alignment, where each step employs a simple deformation to progressively improve the alignment result. The nice thing is that simple deformation can be nicely encoded using a neural network rather than the sophisticated deformation between a pair of images. The paper realizes this goal by designing a recurrent neural network where the current deformation is constantly fit as input into the neural network for improvement. In addition, the paper introduces a weakly supervised loss for training the recurrent neural network. The experimental results justify the usefulness of the approach. On the downside, I would like to see a direct comparison between this approach and non-rigid image alignment. At least for the examples shown in the paper, we can use SIFT/FC7 to guide a (grid-based) deformation field that aligns the two images. In addition, it is unclear how to handle partial similarities. Is it possible to modify the network or the loss function to focus on a partial region? Overall, I recommend accepting the paper based on the novelty of the approach. Minor comments: 1) Please articulate the motivation of the paper. 2) Discuss the limitations. Can you handle objects under drastically different viewpoints? 3) The notations are very involved. Please consider using a table to summarize the meaning of all the notations. Post rebuttal: The rebuttal addresses some of my concerns. However, I would not change my score. I was hoping for a comparison with non-deep learning based image alignment technique, e.g., those used in aligning medical images.